# Binocular treatment for amblyopia: A meta-analysis of randomized clinical trials

**Matilde Roda**[1☺]*, **Marco Pellegrini**[1☺], **Natalie Di Geronimo**[1], **Aldo Vagge**[2], **Michela Fresina**[1], **Costantino Schiavi**[1]

1 Ophthalmology Unit, Azienda Ospedaliero-Universitaria di Bologna IRCCS, Bologna, Italy, 2 University Eye Clinic, DINOGMI, Polyclinic Hospital San Martino IRCCS, Genoa, Italy

☺ These authors contributed equally to this work.
* matilde.roda@hotmail.com

## Abstract

### Background

To date, there is still no consensus regarding the effect of binocular treatment for amblyopia. The purpose of this systematic review and meta-analysis was to summarize the available evidence to determine whether binocular treatment is more effective than patching in children with amblyopia.

### Methods

Four electronic databases (PubMed, Scopus, Web of Science, and Cochrane Central Register of Controlled Trials) were searched for studies that compared binocular treatment and patching in children with amblyopia. The outcome measures were visual acuity and stereopsis. Pooled effects sizes were calculated with a random-effect model. The standardized difference in means (SDM) with 95% confidence intervals (CI) was calculated. Sensitivity analysis and assessment of publication bias were performed.

### Results

Five randomized clinical trials were included. No significant difference in visual acuity between patients treated with binocular treatment and patching was observed (SDM = -0.12; 95% CI: -0.45–0.20; $P$ = 0.464). No significant difference in stereopsis between patients treated with binocular treatment and patching was observed (SDM = -0.07; 95% CI: -0.61–0.48; $P$ = 0.809). For both variables, the between-study heterogeneity was high (respectively, $I^2$ = 61% and $I^2$ = 57%).

### Conclusions

This meta-analysis found no convincing evidence supporting the efficacy of binocular treatment as an alternative to conventional patching. Therefore, the binocular treatment cannot fully replace traditional treatment but, to date, it can be considered a valid complementary therapy in peculiar cases. Further studies are required to determine whether more engaging therapies and new treatment protocols are more effective.

**Data Availability Statement:** All relevant data are within the manuscript and its Supporting Information files.

**Funding:** The authors received no specific funding for this work.

**Competing interests:** The authors have declared that no competing interests exist.

# 1 Introduction

Amblyopia refers to a sensory anomaly characterized by a unilateral or bilateral decrease of visual acuity caused by pattern vision deprivation or abnormal binocular interaction during the critical period of visual development (the first 8–10 years of life) for which no causes can be detected by the physical examination of the eye [1]. It is the most common cause of monocular vision loss in children, occurring in 2% to 5% of them [2, 3]. The most common risk factors for the development of amblyopia are strabismus and anisometropia, or the combination of both [4]. Patching or other means of deprivation of the non-amblyopic eye is the mainstay of amblyopia treatment [5]. The Pediatric Eye Disease Investigator Group (PEDIG) studies have demonstrated that 2–6 hours a day of patching or atropine penalization are both suitable choice for mild and moderate amblyopia [6, 7].

Recently, binocular approaches to treating functional amblyopia have been designed. Hess et al. firstly introduced one of the most promising therapies, namely the anti-suppression, or dichoptic training, in which normal contrast images are presented to the amblyopic eye and low contrast images to the fellow eye [8, 9]. Other strategies include balanced binocular viewing and Interactive Binocular Treatment (I-BiT$^{TM}$), which are performed wearing shutter glasses that present degraded images to the fellow eye in order to balance the input into the visual cortex [10]. The primary goal is to strengthen the amblyopic eye by improving fusion and stereopsis, in order to increase not only visual acuity (VA), but also binocular function. However, controversy remains regarding the efficacy of binocular amblyopia treatment [5].

The purpose of this systematic review and meta-analysis of randomized clinical trials (RCT) was to determine if binocular treatment is comparable, or even superior to patching in treating children with unilateral amblyopia, in order to obtain an improvement in VA and stereoacuity.

# 2 Materials and methods

## 2.1 Search strategy

The meta-analysis was conducted according to the Preferred Reporting Items for Systematic Reviews and Meta-Analyses guidelines [11]. In a priori protocol, we defined eligibility criteria, search strategy, outcomes of interest and analysis methods. An Internet literature search of PubMed, Scopus, Web of Science, and Cochrane Central Register of Controlled Trials databases was performed (last accessed March 25, 2020) to identify studies on binocular therapy for amblyopia. The key words "amblyopia" OR "lazy eye" combined with "binocular amblyopia treatment" OR "binocular amblyopia therapy" OR "binocular therapy" OR "binocular treatment" OR "binocular game" OR "dichoptic" were used in the literature search. No language restriction was applied. In addition, we manually searched the references of selected retrieved articles to identify additional relevant studies.

## 2.2 Eligibility criteria

Published studies were considered eligible if they fulfilled all of the following criteria: study type: RCT; population: patients having anisometropic, strabismic or combined amblyopia aged three to seventeen years; intervention: binocular treatment versus patching; outcome variables: report of at least one among visual acuity and stereoacuity; language: English. Abstracts from conferences, letters, reviews, duplicate publications, full texts without raw data available for retrieval, studies with a paired-eye design and studies in which binocular therapy was combined with other therapies were excluded.

### 2.3 Study selection

After removing duplicate publications, the titles and abstracts of all identified citations were scanned by two independent reviewers (MR and NDG). The full text of citations judged as potentially eligible were obtained and independently screened for eligibility by the same two reviewers (MR and NDG). Disagreements were resolved by discussion with all authors.

### 2.4 Data extraction and risk of bias assessment

The following data were extracted independently from the included trials by two reviewers (MR and NDG) through a pilot-tested data extraction form: first author, year of publication, country, journal, numbers of patients randomized to each arm, 6 study population characteristics (age and type of amblyopia), duration of the study, treatment characteristics, length of treatment, outcome variables (visual acuity and stereoacuity) in each arm. Discrepancies were mediated by a third reviewer (MP). The study quality was assessed using the Revised Cochrane risk-of-bias tool for randomized trials [12]. Two independent reviewers (MR and NDG) assigned a judgment of high, low, or unclear risk of bias for each of the following 5 domains: bias arising from the randomization process, bias due to deviations from intended interventions, bias due to missing outcome data, bias in measurement of the outcome, bias in selection of the reported result.

### 2.5 Data analysis

The meta-analysis was conducted using the Meta package with R (version 4.0.0) and RStudio (version 1.2.5042) software. Mean, standard deviation and sample size were used to calculate the standardized difference in means (SMD) with 95% confidence intervals (CI). $P < 0.05$ was considered statistically significant. The $I^2$ tests was used to evaluate heterogeneity. The results were pooled using a random effect model and the DerSimonian and Laird method [13]. To explore individual study effects on overall pooling, sensitivity analyses were performed removing one study at a time from the analysis [14]. Potential publication bias was assessed by visual evaluation of the funnel plots.

## 3 Results

### 3.1 Literature search

Of the 658 potentially relevant studies identified from electronic databases and hand searches, 24 articles were retrieved for full-text review after adjusting for duplicates and titles and screening of titles and abstracts. After exclusion of 19 studies based on the predefined inclusion criteria, 5 studies were retained for the meta-analysis [15–19]. The flow chart of the study selection is presented in Fig 1.

### 3.2 Characteristics of the studies

Table 1 shows a summary of the characteristics of the included studies. The selected trials were reported between 2016 and 2019. Three studies were conducted in United States, one in Iran, and one in China. They examined the effect of binocular therapy for amblyopia compared to patching. Of 625 involved patients, all presented functional amblyopia, in particular: anisometropic amblyopia (n = 338), strabismic amblyopia (n = 89), combined anisometropic/strabismic amblyopia (n = 160), and unspecified (n = 38). The final visit was completed by 587 patients. The range of average ages was from 3 to 17 years. In all studies binocular therapy consisted in dichoptic training played at home for a range from 20 minutes to 1 hour a day 5–7

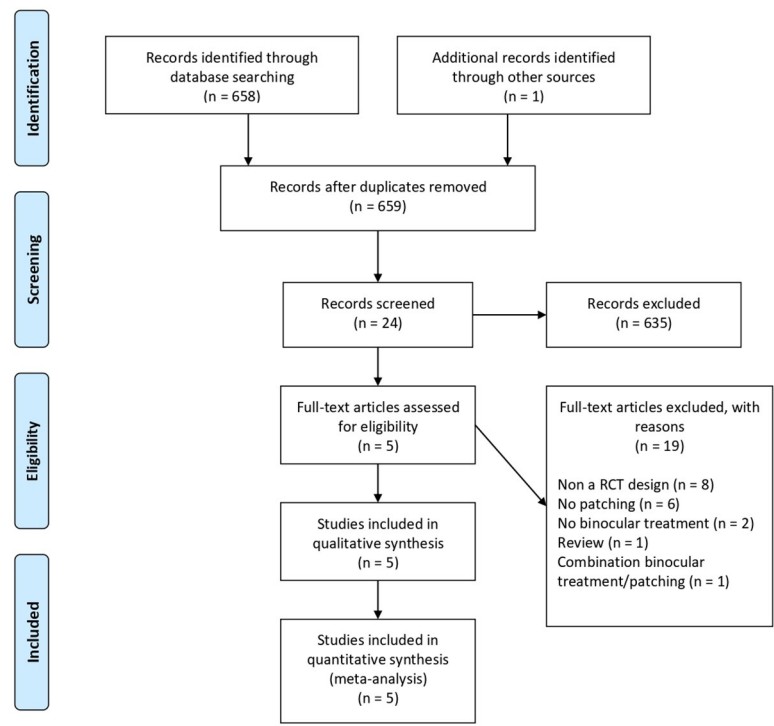

From: Moher D, Liberati A, Tetzlaff J, Altman DG, The PRISMA Group (2009). *Preferred Reporting Items for Systematic Reviews and Meta-Analyses: The PRISMA Statement. PLoS Med 6(7): e1000097. doi:10.1371/journal.pmed1000097*

For more information, visit www.prisma-statement.org.

**Fig 1. PRISMA flow-chart.**

**Table 1. Characteristics of the randomized clinical trials included in the meta-analysis.**

| Author (year) | Cause of Amblyopia (n) | Sample Size Total/BinT/ Patch | Length of Treatment | Age (median, yrs) | BinT Regimen | Patch Regimen |
|---|---|---|---|---|---|---|
| Holmes et al. (2016) | · A (199) <br> · S (66) <br> · A/S (120) | 385/190/195 | 4 months | · BinT: 8.6 <br> · Patch: 8.4 | · iPad game <br> · 1 hr/day | · Patching <br> · 2 hrs/day |
| Kelly et al. (2016) | · A (14) <br> · S (9) <br> · A/S (5) | 28/14/14 | 2 weeks | · BinT: 6.6 <br> · Patch: 6.95 | · Action adventure game <br> · 1 hr/day <br> · 5 days/wk | · Patching <br> · 2 hrs/day |
| Mahn et al. (2018) | · A (51) <br> · S (14) <br> · A/S (35) | 100/40/60 | 4 months | · BinT: 14.3 <br> · Patch: 14.3 | · Falling blocks game <br> · 1 hr/day | · Patching <br> · 2 hrs/day |
| Rajavi et al. (2019) | Non specified | 38/17/21 | 1 month | · BinT: 6.5 <br> · Patch: 7.55 | · I-BiT game <br> · 20 min/session <br> · 5 sessions/wk | · Patching <br> · 2–4 hrs/day + Placebo IBiT |
| Yao et al. (2018) | A (76) | 76/36/38 | 3 months | · BinT: 5.92 <br> · Patch: 5.68 | · Binocular game <br> · 40 min/day | · Patching <br> · 2–6 hrs/day |

A, Anisometropic amblyopia; S, Strabismic amblyopia; A/S, combined anisometropic/strabismic amblyopia; BinT, Binocular treatment group; Patch: Patching group

**Table 2. Quality of the studies included in the meta-analysis assessed by revised Cochrane risk-of-bias tool for randomized trials (RoB 2).**

| Author (year) | Randomization process | Deviations from intended interventions | Missing outcome data | Measurements of the outcome | Selection of the reported result | Overall |
|---|---|---|---|---|---|---|
| Holmes et al. (2016) | ? | ? | + | + | + | ! |
| Kelly et al. (2016) | + | ? | + | + | + | ! |
| Mahn et al. (2018) | + | ? | + | ? | + | ! |
| Rajavi et al. (2019) | ? | + | + | ? | + | ! |
| Yao et al. (2018) | + | ? | + | + | + | ! |

+, Low risk;?, Some concerns; -, Low risk.

days a week; patching was prescribed 2–6 hours/day. In one study [18] patching was associated with a placebo binocular game. The length of treatment ranged from 2 weeks to 4 months.

Compliance was lower in the binocular therapy group compared to the control group in Holmes's study (67% versus 92%) [15], in Mahn's study (62% versus 75%) [17] and Yao study (85% versus 90%) [19]. Conversely, compliance was higher in the binocular therapy group in Rajavi's study (87.5% versus 76%) [18], whereas it was excellent in both groups in Kelly's study (100% versus 99%) [16].

### 3.3 Quality assessment results

The risk of bias assessment for each study included in the meta-analysis is shown in Table 2. All studies are free of risk of bias from missing outcome data and selection of the reported results, three (60%) from randomization process and measurement of the outcome, and only one (20%) from deviations from intended interventions. No study presented high risk of bias for any domain. Overall, all studies showed some concerns about risk of bias.

### 3.4 Quantitative analysis, sensitivity analysis and publication bias

Best corrected visual acuity was reported in all the studies, and was measured in logMAR in 4 of them [15, 16, 18, 19], and in number of read letters at E-ETDRS charts in another [17]. The pooled SDM of VA was -0.12 (95% CI: -0.45–0.20; $P = 0.464$), indicating no significant difference in VA between patients treated with binocular treatment versus patching (Fig 2). The

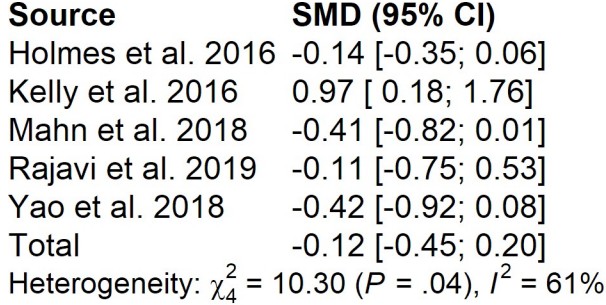
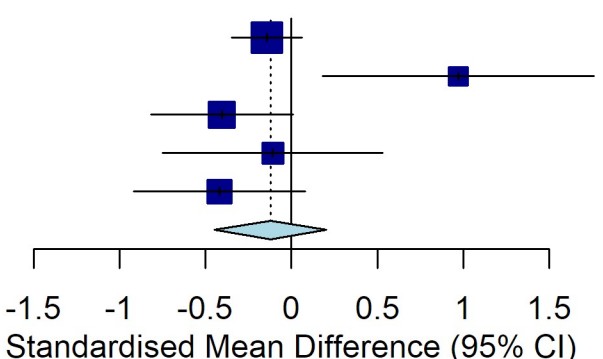

**Fig 2. Difference in VA between patients treated with binocular treatment versus patching.**

| Source | SMD (95% CI) |
|---|---|
| Kelly et al. 2016 | 0.00 [-0.74; 0.74] |
| Rajavi et al. 2019 | -0.59 [-1.25; 0.06] |
| Yao et al. 2018 | 0.31 [-0.19; 0.80] |
| Total | -0.07 [-0.61; 0.48] |
| Heterogeneity: $\chi_2^2 = 4.60$ ($P = .10$), $I^2 = 57\%$ | |

**Fig 3. Difference in stereopsis between patients treated with binocular treatment versus patching.**

between-study heterogeneity was high (Q = 10.3, $I^2$ = 61%). The results were not stable following sensitivity analysis; in particular, excluding the study of Kelly et al. (16), the SDM was -0.21 (95% CI: -0.38–0.05; $P$ = 0.012), indicating a higher improvement in VA in patients treated with patching. No significant publication bias was detected by visual evaluation of the funnel plot.

Stereopsis was retrieved from 3 studies, and was tested with the Randot Preschool stereoacuity tests in one of them [16], and with Titmus stereo test in two of them [18, 19]. The pooled SDM of stereoacuity was -0.07 (95% CI: -0.61–0.48; P = 0.809), indicating no significant difference in stereopsis between patients treated with binocular treatment versus patching (Fig 3). The between-study heterogeneity was high (Q = 4.6, $I^2$ = 57%). The sensitivity analysis performed by excluding any single study at a time from the meta-analysis did not significantly change the pooled SDM, which ranged from -0.32 (95% CI: -0.90–0.26; $P$ = 0.274) to 0.21 (95% CI: -0.20–0.62; $P$ = 0.315). No significant publication bias was detected by visual evaluation of the funnel plot.

## 4 Discussion

To date, occlusion therapy with patching is still the mainstay of amblyopia treatment in children and adolescents [6]. During the years, many studies investigated the safety and effectiveness of novel strategies targeted to non-compliance patients, and binocular therapies have recently received widespread attention [5]. Several prospective studies on binocular treatment were promising [10], but most recent RCTs showed contrasting results [5].

This meta-analysis of 5 RCTs aimed at investigating the efficacy of new binocular dichoptic therapies versus patching in the treatment of amblyopia. The qualitative analysis underlined some important differences among the studies. First, all studies included patients during the critical period of visual development (the first 8–10 years of life), except the one by Manh et al. [17], which included adolescents between 13 and 17 years. Although adults conserve some forms of synaptic plasticity, the mechanism for the induction and expression of plasticity differs from those used during the critical period [20]. Second, three studies included patients with anisometropic amblyopia, strabismic amblyopia or a combination of both [15–17], one study only with anisometropic amblyopia [19], while another did not specify the cause of amblyopia [18]. The impact of amblyopia subtype on binocular treatment success is still debated [5], and the lack of a perfect eyes' alignment in strabismic amblyopia might influence the binocular therapy outcomes. Third, each trial used different binocular games and treatment protocol, whereas the protocols for patching were more similar among studies.

The quantitative analysis showed no superiority of binocular therapy over patching in terms of visual acuity and stereopsis. In details, four out of five studies reported no significant difference between binocular therapy and patching in improving VA [15, 17–19]. Conversely,

the VA improvement was higher in patients receiving binocular therapy in the study from Kelly and colleagues [16]. This difference could be due to the shorter trial duration (only two weeks). In fact, patching usually requires a longer treatment duration for being effective [21]. The sensitivity analysis performed by excluding the study by Kelly et al. [16] demonstrated a higher improvement in VA in patients receiving patching. Conversely, the stereopsis results were stable following the sensitivity analysis.

Compliance represents one of the most important factors affecting the success of amblyopia treatment [22]. Previously, binocular therapy was supposed to be an effective solution for those with poor adherence to treatment [5]. However, in three of the studies included in this meta-analysis, compliance appeared better in patients treated with patching [15, 17, 19]. Conversely, it was higher in patients treated with binocular game in one study [18], and was excellent in both groups in another [16]. Furthermore, in two studies using the binocular therapy devices to check the compliance to treatment, the objectively recorded compliance was even lower compared to what reported by parents [15, 17]. However, the PEDIG group hypothesized that higher compliance for the binocular therapy could be obtained in the future with more engaging games [15], and this might result in higher efficacy of the treatment.

In 2015, a Cochrane Review reported no published controlled trials on binocular treatment [10]. Subsequently, RCTs were completed, and the recent report by the American Academy of Ophthalmology [5] concluded that to date there is no evidence to substitute standard therapies of amblyopia with binocular strategies. Some results were published by Brin et al. [23]. Moreover, Chen et al. recently compared patching and binocular treatment reporting that patching is statistically better than binocular treatment [24].

Our meta-analysis including all the published RCTs comparing binocular therapy and patching is in line with this report and confirms that binocular therapy cannot fully replace traditional treatment but, to date, it can be considered a valid complementary therapy in peculiar cases.

The principal limitation of this meta-analysis is the high heterogeneity in effect estimation. This may be due to the above-mentioned differences among studies, such as age of patients, type of amblyopia, duration, daily dose and type of binocular therapy, all of which might influence the treatment efficacy. Moreover, the RCTs used different tests and measurement methods for VA and stereopsis. The discrepancy in outcome measures is a recognized issue in amblyopia RCTs [25], and it might have contributed to the inconsistency among studies. Finally, besides the outcome variables of this meta-analysis, other factors such as cost and availability should be considered when selecting the most appropriate amblyopia therapy in an individual patient.

In conclusion, this meta-analysis found no convincing evidence to support the superiority of binocular therapy compared to conventional patching. Therefore, the use of binocular therapy can just be considered as a supplementary strategy to associate to the traditional patching treatment. However, further studies might be useful to explore whether more engaging therapies and new treatment protocols yield better results.

## Supporting information

**S1 Checklist. PRISMA 2009 checklist.**
(DOC)

## Author Contributions

**Conceptualization:** Matilde Roda, Marco Pellegrini, Natalie Di Geronimo, Aldo Vagge, Costantino Schiavi.

**Data curation:** Matilde Roda, Marco Pellegrini, Natalie Di Geronimo.

**Formal analysis:** Marco Pellegrini.

**Investigation:** Matilde Roda.

**Methodology:** Matilde Roda.

**Project administration:** Matilde Roda.

**Supervision:** Costantino Schiavi.

**Validation:** Matilde Roda, Natalie Di Geronimo, Aldo Vagge, Michela Fresina, Costantino Schiavi.

**Visualization:** Aldo Vagge, Michela Fresina.

**Writing – original draft:** Matilde Roda, Natalie Di Geronimo.

**Writing – review & editing:** Matilde Roda, Aldo Vagge, Michela Fresina, Costantino Schiavi.

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
