## [Decision Letter · Decision Letter 0]

26 May 2021

PONE-D-21-02836

Binocular treatment for amblyopia: a meta-analysis of randomized clinical trials

PLOS ONE

Dear Dr. Roda,

Thank you for submitting your manuscript to PLOS ONE. After careful consideration, we feel that it has merit but does not fully meet PLOS ONE’s publication criteria as it currently stands. Therefore, we invite you to submit a revised version of the manuscript that addresses the points raised during the review process.

Both reviewers have raised concerned about the final conclusions that the binocular therapy is not recommended. This could be made rather subtle. As most studies have suggested that binocular therapy could be a potential treatment alternative. Besides, as highlighted by the Reviewer-1, the contrast between this study and the other recent meta-analysis from Chen et al. is worth discussing. 

We look forward to receiving your revised manuscript.

Kind regards,

Arijit Chakraborty, PhD

Academic Editor

PLOS ONE

Journal Requirements:

2. Please include your tables as part of your main manuscript and remove the individual files. Please note that supplementary tables should remain uploaded as separate "supporting information" files.

3. As per PLOS' guidelines (https://journals.plos.org/plosone/s/submission-guidelines#loc-systematic-reviews-and-meta-analyses), please ensure that you have provided details regarding record exclusion for all steps of the screening process in Fig 1.

4. Please also ensure that you have identified the study as a both a systematic review and meta-analysis in the title.

Reviewers' comments:

Reviewer's Responses to Questions

**Comments to the Author**

1. Is the manuscript technically sound, and do the data support the conclusions?

Reviewer #1: Yes

Reviewer #2: Partly

2. Has the statistical analysis been performed appropriately and rigorously? 

Reviewer #1: Yes

Reviewer #2: Yes

3. Have the authors made all data underlying the findings in their manuscript fully available?

Reviewer #1: Yes

Reviewer #2: Yes

4. Is the manuscript presented in an intelligible fashion and written in standard English?

Reviewer #1: Yes

Reviewer #2: Yes

5. Review Comments to the Author

Reviewer #1: The authors of the study performed meta-analysis to compare the effectiveness of newly emerging binocular treatment and the gold-standard patching therapy for the management of amblyopia. For this analysis, they used 5 RCTs that compared the binocular treatment and patching therapy (Holmes et al, 2016; Kelly et al. 2018; Mahn et al. 2018; Yao et al. 2018 & Rajavi et al. 2019). The results showed that there is no compelling evidence to show that binocular therapy is superior to the conventional patching therapy in terms of improving visual acuity and stereoacuity. Hence the authors concluded that the binocular therapy is “not recommended” for the management of amblyopia. This meta-analysis is very well planned and analyzed. This is very important for clinicians and scientists to understand the effectiveness of newly emerging binocular treatment of amblyopia. The followings are my comments and suggestions to the authors of this manuscript.

This is an important study and full credits to the authors for this detailed and comprehensive meta-analysis. However, I recently came across another study that has done meta-analysis to compare the effectiveness of binocular treatment and patching therapy for the management of amblyopia (Chen et al. 2021, BMJ Open Ophthalmol. 2021; 6(1): e000625). This article was published in February 2021, so there is a high chance that the authors of the current paper would have not noticed the previous paper before conducting the same meta-analysis.

While comparing the results of these two meta-analysis papers, there seems to be no consensus. The analysis by Chen et al. showed that best corrected visual acuity was significantly improved by binocular treatment compared to patching therapy but the analysis of the current paper suggested otherwise – though both analyses used results from the same set of RCTs. This discrepancy between these two analyses should be addressed in the current paper.

Another discrepancy is number of RCTs (n=5) used in this analysis compared to the study by Chen et al. (n=6). On a quick perusal, it seems like both studies used the same exclusion/inclusion criteria. The authors of the current paper should address, why did they exclude the study by Rajavi et al. 2018? Moreover, just curious whether is this the reason for a discrepancy in the results of the analysis in terms of BCVA?

Finally, no significant difference in the improvement of visual acuity between binocular therapy and conventional patching therapy means that the former method is as efficient as the latter. Moreover, supervised dichoptic training under laboratory setting (Li et al., 2013) and binocular treatment with higher compliance (Kelly et al., 2019) suggest that binocular treatment can be superior to patching if compliance is achieved. Therefore, strong criticism such as “not recommended” based on the results of meta-analysis can be avoided – instead, both treatment options can be provided to patients and let the patient or parent/guardian of the patient to decide. Nonetheless, I agree with authors’ recommendation that more RCTs are needed to properly estimate the efficacy of binocular treatment of amblyopia.

Reviewer #2: Roda et al. provide a structured review focussing on RCTs that compare binocular amblyopia treatment to patching. I enjoyed reading the paper and the methodology appears sound. My comments follow:

1) The final paragraph of the introduction is rather vague. Please specify the exact question addressed by the review. Presumably it involves a comparison between binocular treatment and patching on VA and/or stereo in children with amblyopia.

2) Methods: the literature search was conducted over 1 year ago. This should be updated and any new RCTs added to the manuscript.

3) A greater discussion of the specifics of each binocular treatment included is required. For example, some the techniques require both eyes to be used simultaneously in a complementary manner whereas others induce competition between the two eyes. Fundamental differences in treatment approaches such as these make meta-analyses challenging. This point should be highlighted.

4) Section 2.4. How is logMAR different from E-ETDRS letters? One is a measurement unit and the other an optotype. Please clarify

5) Discussion paragraph 4. There is repetition from previous sections here. It would be good to consolidate the discussion of adherence into a single section.

6) In the abstract and at the end of the discussion the authors conclude that binocular treatment cannot be recommended. However, the main analyses don't show that binocular treatment is worse than patching, only that it is no different which could make it an acceptable treatment alternative. A conclusion that is more directly supported by the meta-analysis results, particularly the high heterogeneity, is that current studies are inconclusive and variable adherence is a confounding factor.

6. PLOS authors have the option to publish the peer review history of their article (what does this mean?). If published, this will include your full peer review and any attached files.

Reviewer #1: **Yes: **Rajkumar Nallour Raveendran

Reviewer #2: No

---

## [Author Response · Author response to Decision Letter 0]

10 Jun 2021

First of all, we would like to thank both the reviewers for appreciating our paper.

This is our reply to comments:

Reviewer #1: The authors of the study performed meta-analysis to compare the effectiveness of newly emerging binocular treatment and the gold-standard patching therapy for the management of amblyopia. For this analysis, they used 5 RCTs that compared the binocular treatment and patching therapy (Holmes et al, 2016; Kelly et al. 2018; Mahn et al. 2018; Yao et al. 2018 & Rajavi et al. 2019). The results showed that there is no compelling evidence to show that binocular therapy is superior to the conventional patching therapy in terms of improving visual acuity and stereoacuity. Hence the authors concluded that the binocular therapy is “not recommended” for the management of amblyopia. This meta-analysis is very well planned and analyzed. This is very important for clinicians and scientists to understand the effectiveness of newly emerging binocular treatment of amblyopia. The followings are my comments and suggestions to the authors of this manuscript.

This is an important study and full credits to the authors for this detailed and comprehensive meta-analysis. However, I recently came across another study that has done meta-analysis to compare the effectiveness of binocular treatment and patching therapy for the management of amblyopia (Chen et al. 2021, BMJ Open Ophthalmol. 2021; 6(1): e000625). This article was published in February 2021, so there is a high chance that the authors of the current paper would have not noticed the previous paper before conducting the same meta-analysis.

While comparing the results of these two meta-analysis papers, there seems to be no consensus. The analysis by Chen et al. showed that best corrected visual acuity was significantly improved by binocular treatment compared to patching therapy but the analysis of the current paper suggested otherwise – though both analyses used results from the same set of RCTs. This discrepancy between these two analyses should be addressed in the current paper.

Thanks for your comment. We actually notice the recent publication of Chen et al’s paper, who performed a research very similar to ours. We submitted our meta-analysis at the end of January before the publication by Chen et al., and we believe that our study is more reliable because Chen et al. included the study by Rajavi et al 2016 which compared patching and combination of patching and i-Bit. Moreover, they concluded that patching is statistically better than binocular treatment, but it is not correct. We observed that there is no significant difference in visual acuity between patients treated with binocular treatment and patching. 

Another discrepancy is number of RCTs (n=5) used in this analysis compared to the study by Chen et al. (n=6). On a quick perusal, it seems like both studies used the same exclusion/inclusion criteria. The authors of the current paper should address, why did they exclude the study by Rajavi et al. 2018? Moreover, just curious whether is this the reason for a discrepancy in the results of the analysis in terms of BCVA?

When we performed the review, we decided on purpose to exclude Rajavi and colleagues’ paper (Rajavi Z, Sabbaghi H, Amini Sharifi E, et al. The role of interactive binocular treatment system in amblyopia therapy. J Curr Ophthalmol 2016;28:217–22) . Indeed, their trial compared a case group, that underwent a combined therapy with i-Bit games and patching, to a control group which was treated only with patch therapy. We believe that was inappropriate and unreliable to compare this protocol of treatment to those that were applied by the other authors, where children underwent only one type of treatment at a time. Moreover, the main purpose of our study was to compare the actual effectiveness of each treatment, and not the combination of both, as Rajavi et collegues did. 

Finally, no significant difference in the improvement of visual acuity between binocular therapy and conventional patching therapy means that the former method is as efficient as the latter. Moreover, supervised dichoptic training under laboratory setting (Li et al., 2013) and binocular treatment with higher compliance (Kelly et al., 2019) suggest that binocular treatment can be superior to patching if compliance is achieved. Therefore, strong criticism such as “not recommended” based on the results of meta-analysis can be avoided – instead, both treatment options can be provided to patients and let the patient or parent/guardian of the patient to decide. Nonetheless, I agree with authors’ recommendation that more RCTs are needed to properly estimate the efficacy of binocular treatment of amblyopia.

We thank you for your observation. Actually, binocular treatment is a promising approach that deserve more attention and diffusion; therefore, we will revise our conclusion from binocular treatment as “not recommended” to “a reliable alternative or complementary approach”.

Reviewer #2: Roda et al. provide a structured review focussing on RCTs that compare binocular amblyopia treatment to patching. I enjoyed reading the paper and the methodology appears sound. My comments follow:

• The final paragraph of the introduction is rather vague. Please specify the exact question addressed by the review. Presumably it involves a comparison between binocular treatment and patching on VA and/or stereo in children with amblyopia.

Thanks for the correction. We will clarify better the purpose of our study.

• Methods: the literature search was conducted over 1 year ago. This should be updated and any new RCTs added to the manuscript.

We performed our research about a year ago, but in the last months any other trial suitable for our metanalysis has been published, hence there aren’t usable new data to add to the review.

• A greater discussion of the specifics of each binocular treatment included is required. For example, some the techniques require both eyes to be used simultaneously in a complementary manner whereas others induce competition between the two eyes. Fundamental differences in treatment approaches such as these make meta-analyses challenging. This point should be highlighted.

It was quite difficult for us to to deepen the specific of each binocular treatment since the authors didn’t give precise information about the peculiar systems they employed. In the second paragraph of the discussion, we underline the main differences between the RCTs, which include the treatment protocols as well. A greater discussion regarding the multiple and specific existing methods to adopt binocular treatment fall outside our essay.

• Section 2.4. How is logMAR different from E-ETDRS letters? One is a measurement unit and the other an optotype. Please clarify

Thank for noticing the error. We will explaine more clearly the methods authors used to estimate BCVA.

• Discussion paragraph 4. There is repetition from previous sections here. It would be good to consolidate the discussion of adherence into a single section.

Thanks for the comment. We don’t agree with your observation because we talked about compliance only in the paragraph 4. 

• In the abstract and at the end of the discussion the authors conclude that binocular treatment cannot be recommended. However, the main analyses don't show that binocular treatment is worse than patching, only that it is no different which could make it an acceptable treatment alternative. A conclusion that is more directly supported by the meta-analysis results, particularly the high heterogeneity, is that current studies are inconclusive and variable adherence is a confounding factor.

Thank you very much for the revision. We will revise our conclusion in order to be more supportive to this new therapeutical approach.

---

## [Decision Letter · Decision Letter 1]

16 Jul 2021

PONE-D-21-02836R1

Binocular treatment for amblyopia: a meta-analysis of randomized clinical trials

PLOS ONE

Dear Dr. Roda,

Thank you for revising your manuscript to PLOS ONE. While both reviewers are overall satisfied with the current version of the manuscript, the reviewer-2 has raised some additional concerns. Those are fair concerns and therefore, we invite you to submit a revised version of the manuscript that addresses the points raised by the reviewer-2. 

We look forward to receiving your revised manuscript.

Kind regards,

Arijit Chakraborty, PhD

Academic Editor

PLOS ONE

Journal Requirements:

Reviewers' comments:

Reviewer's Responses to Questions

**Comments to the Author**

1. If the authors have adequately addressed your comments raised in a previous round of review and you feel that this manuscript is now acceptable for publication, you may indicate that here to bypass the “Comments to the Author” section, enter your conflict of interest statement in the “Confidential to Editor” section, and submit your "Accept" recommendation.

Reviewer #1: All comments have been addressed

Reviewer #2: (No Response)

2. Is the manuscript technically sound, and do the data support the conclusions?

Reviewer #1: Yes

Reviewer #2: Yes

3. Has the statistical analysis been performed appropriately and rigorously? 

Reviewer #1: Yes

Reviewer #2: Yes

4. Have the authors made all data underlying the findings in their manuscript fully available?

Reviewer #1: Yes

Reviewer #2: Yes

5. Is the manuscript presented in an intelligible fashion and written in standard English?

Reviewer #1: Yes

Reviewer #2: Yes

6. Review Comments to the Author

Reviewer #1: (No Response)

Reviewer #2: The authors have addressed most of my comments appropriately. However, I agree with Reviewer 1 that a comparison with the Chen study is required in the discussion section. The authors should also comment on another recent review by Brin et al. that draws the same conclusions that they do. https://bmjophth.bmj.com/content/6/1/e000657.

Specific comments:

1. A paragraph should be added to the discussion comparing this study to other recently published meta analyses in the field. This will help to place the study in the context of the current literature.

2. Newly added text in the abstract and discussion should be proof read for grammar and word choice.

7. PLOS authors have the option to publish the peer review history of their article (what does this mean?). If published, this will include your full peer review and any attached files.

Reviewer #1: **Yes: **RAJKUMAR NALLOUR RAVEENDRAN

Reviewer #2: No

---

## [Author Response · Author response to Decision Letter 1]

18 Jul 2021

We appreciate the Editorial Board Member and the Reviewers for the opportunity to revise our work for consideration for publication on PlosOne. 

This is the reply to Reviewer #2:

1. A paragraph should be added to the discussion comparing this study to other recently published meta analyses in the field. This will help to place the study in the context of the current literature.

We added in the discussion paragraph.

2. Newly added text in the abstract and discussion should be proof read for grammar and word choice.

Ok, we did.

---

## [Decision Letter · Decision Letter 2]

16 Sep 2021

Binocular treatment for amblyopia: a meta-analysis of randomized clinical trials

PONE-D-21-02836R2

Dear Dr. Roda,

We’re pleased to inform you that your manuscript has been judged scientifically suitable for publication and will be formally accepted for publication once it meets all outstanding technical requirements.

Kind regards,

Arijit Chakraborty, PhD

Academic Editor

PLOS ONE

Additional Editor Comments (optional):

Reviewers' comments:

Reviewer's Responses to Questions

**Comments to the Author**

1. If the authors have adequately addressed your comments raised in a previous round of review and you feel that this manuscript is now acceptable for publication, you may indicate that here to bypass the “Comments to the Author” section, enter your conflict of interest statement in the “Confidential to Editor” section, and submit your "Accept" recommendation.

Reviewer #2: All comments have been addressed

2. Is the manuscript technically sound, and do the data support the conclusions?

Reviewer #2: (No Response)

3. Has the statistical analysis been performed appropriately and rigorously? 

Reviewer #2: (No Response)

4. Have the authors made all data underlying the findings in their manuscript fully available?

Reviewer #2: (No Response)

5. Is the manuscript presented in an intelligible fashion and written in standard English?

Reviewer #2: (No Response)

6. Review Comments to the Author

Reviewer #2: (No Response)

7. PLOS authors have the option to publish the peer review history of their article (what does this mean?). If published, this will include your full peer review and any attached files.

Reviewer #2: No

---

## [Editor Report · Acceptance letter]

22 Sep 2021

PONE-D-21-02836R2 

Binocular treatment for amblyopia: a meta-analysis of randomized clinical trials 

Dear Dr. Roda:

I'm pleased to inform you that your manuscript has been deemed suitable for publication in PLOS ONE. Congratulations! Your manuscript is now with our production department. 

Kind regards, 

on behalf of

Dr. Arijit Chakraborty 

Academic Editor

PLOS ONE